# Study of Ultra-Broadband Synthesizer of Fast Indirect Type in a 0.5–18 GHz Range for SIGINT System

**Yuseok Jeon**

R&D Department, Broadern Inc., 32-5, Dongtangiheungro, Hwaseong-si 18487, Republic of Korea; ysjeon@broadern.co.kr

**Abstract:** In the present study, a new structure (0.5–18 GHz) with excellent phase noise characteristics and a fast switching speed is proposed. Ultra-wideband synthesizers with low phase noises and less spurious signals have been developed to be used as local oscillators and for built-in test (BIT) functions in the field of electronic warfare systems (EW), in which synthesizers are installed at the front-end of devices; this is accomplished by applying an SMT process using a packaged-type device. This paper compares the advantages and disadvantages of PLVCO (old) and DDS reference sources (new) based on two types of frequency synthesizers. At this time, different frequencies can be output from the two ports by different FPGA coding of the internal frequency plan according to the operating frequency. The main RF line will be made a dielectric substrate, RO4350, with a relative dielectric constant of 3.38 and a dielectric thickness of 0.508 mm. In the ultra-broadband synthesizer module, the phase noise of the DDS output (1.25 GHz) was −131 dBc/Hz at 10 KHz offset. The phase noise in the 18 GHz output is expected to be −105.9 dBc/Hz at 1 KHz offset. In particular, by proposing a structure for obtaining a wideband frequency using a single source (DDS), the structure secures reliability from the point of view of a system operating for a long time by implementing a similar circuit within a predictable range.

**Keywords:** front-end; synthesizer; ultra-wideband frequency; phase noise; local circuit; built-in test; low spurious

## 1. Introduction

A frequency synthesizer is an electronic system for generating any of a range of frequencies from a single fixed time base or oscillator. They are found in many modern devices, including radio receivers, mobile telephones, radio telephones, satellite receivers, GPS systems, etc. A frequency synthesizer can combine frequency multiplication, frequency division, and frequency mixing operations to produce the desired output frequency [1]. Over the last few years, because of the many microwave/RF and wireless activities, interest in frequency synthesizers has grown rapidly. Synthesizers are found in test and measurement equipment, as well as in communication equipment [2].

Frequency synthesizers have been well studied, but difficult problems sometimes arise in their practical implementations. The major concern of synthesizer designers is the phase noise [3]. The frequency synthesizer must cover a wide frequency range. A frequency synthesizer avoids the need for using many independent crystal-controlled oscillators in a wideband multiple-channel system. Modern frequency synthesizers can be implemented using integrated circuit chips. They are controlled by digital circuits or computers. Frequency synthesizers are commonly used in transmitters, modulators, and LOs in many wireless communication systems such as radios, satellite receivers, cellular telephones, and data transmission equipment [4]. There are three major frequency synthesis techniques: direct analog, direct digital frequency synthesizer (DDFS), and indirect or phase-locked loop (PLL) [5]. Direct synthesizers are usually used in agile frequency and high improve factor radar systems because of their fast agile time, low phase noise, and

low spurious signal emissions [6]. Direct digital synthesizers (DDS) have pros and cons, as follow:

1.  Advantage

    An almost arbitrarily small channel step size is achievable and extremely fast phase-continuous frequency switching is possible.

2.  Disadvantage

    Performance is ultimately limited by the D/A converter and unless an extremely high FCLK is used, the output must generally be either multiplied or heterodyned to higher frequencies before use [7].

    Frequency multipliers multiply phase noise by the multiplication ratio M, increasing it by $20 \times \log_{10}M$. Frequency dividers divide phase noise by the divide ratio N, changing it by $-20 \times \log_{10}N$ [8]. For military applications, indirect synthesizers can be used for electronic warfare (EW) systems such as radar warning receivers (RWRs), electronic support measurement (ESM), and signal intelligence (SIGINT). The implementation of this type of synthesizer could be both as the local oscillator of the system and as the source for the BIT subsystem [9].

    The super heterodyne (SH) receiver shown in Figure 1 is the most commonly used receiver in communications and radar equipment for its high sensitivity and frequency selectivity. In EW applications, narrow-band (NB) SHs are used primarily in electronic intelligence (ELINT) equipment to isolate an emitter signal from the environment and measure its fine-grain information without any interfering signals, whereas large-band (LB) scanning SHs are used to search for and detect threat emitters in highly dense pulse scenarios because their selectivity strongly reduces the received pulse density. In this architecture, the overall RF bandwidth is divided into a number of sub-bands that are individually shifted into a fixed IF bandwidth by one of a number of local oscillator (LO) frequencies, as generated by a synthesizer or a comb generator that can sweep the overall RF bandwidth [10]. In this paper, we are trying to find the structure and design method of an ultra-wideband synthesizer with a 0.5–18 GHz output using the indirect method. The contents proposed in this paper are as follows.

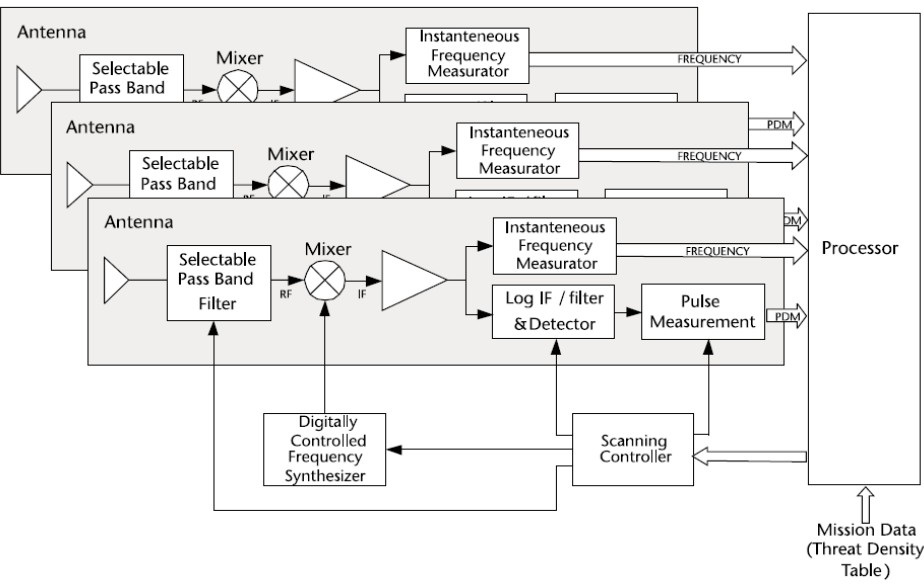

**Figure 1.** Schematic block diagram of a LB SH.

First, in order to output the frequency for LO for down-converting the wideband (0.5–18 GHz) signal received through the antenna and the frequency for BIT for signal correction of the receiver module, both paths are designed to be implemented in the module. Second, since the wideband output frequency of the frequency synthesizer has to match the

operating system and phase synchronization, the 100 MHz signal, which is the reference clock signal for driving the DDS, is used as an external REF. It is designed so that it can be received on the port. Third, a multiplier circuit is designed to generate a clock signal up to 3 GHz in order to output the DDS chip, which is essential to satisfy the fast switching time (settling time) up to a high frequency. When selecting a multiplier method, a passive multiplier method is adopted to prevent deterioration of phase noise characteristics due to the physical property limit of SRD. Fourth, as the frequency is subdivided through multiple paths, the structural design is made to mechanically secure the degree of isolation between channels. Fifth, a tunable filter is used so that the frequency can be varied within the filter bandwidth in association with the DDS output frequency. This is to overcome the disadvantage of using multiple filters when using fixed filters.

## 2. Understanding of Wideband Frequency Synthesizer for 6–18 GHz Range Based on PLVCO (Old Version)

In this study, Figure 2 is a frequency synthesizer with a typical structure. In order to output 6–18 GHz from a single port, it has a 1 GHz reference source for driving DDS, three DDS chips, and three different VCOs and PLLICs. In addition, multipliers and filters were placed on each VCO output to obtain a high frequency at the final output. Since this structure is a form in which each VCO is phase-locked to the DDS output frequency, the in-band phase noise characteristic is poor due to the limit of the closed loop phase noise of the PLLIC. In addition, three different VCO output frequencies serving as different sources may cause signal interference to other paths within a limited structure. Figure 2 shows a block diagram of the design with three VCOs and a multiplier path. This configuration does not generate ultra-wideband frequencies for some sections (6–18 GHz), and if it needs to be extended to 0.5 GHz, it is difficult to place in a limited space due to the increase in additional VCOs and additional circuits.

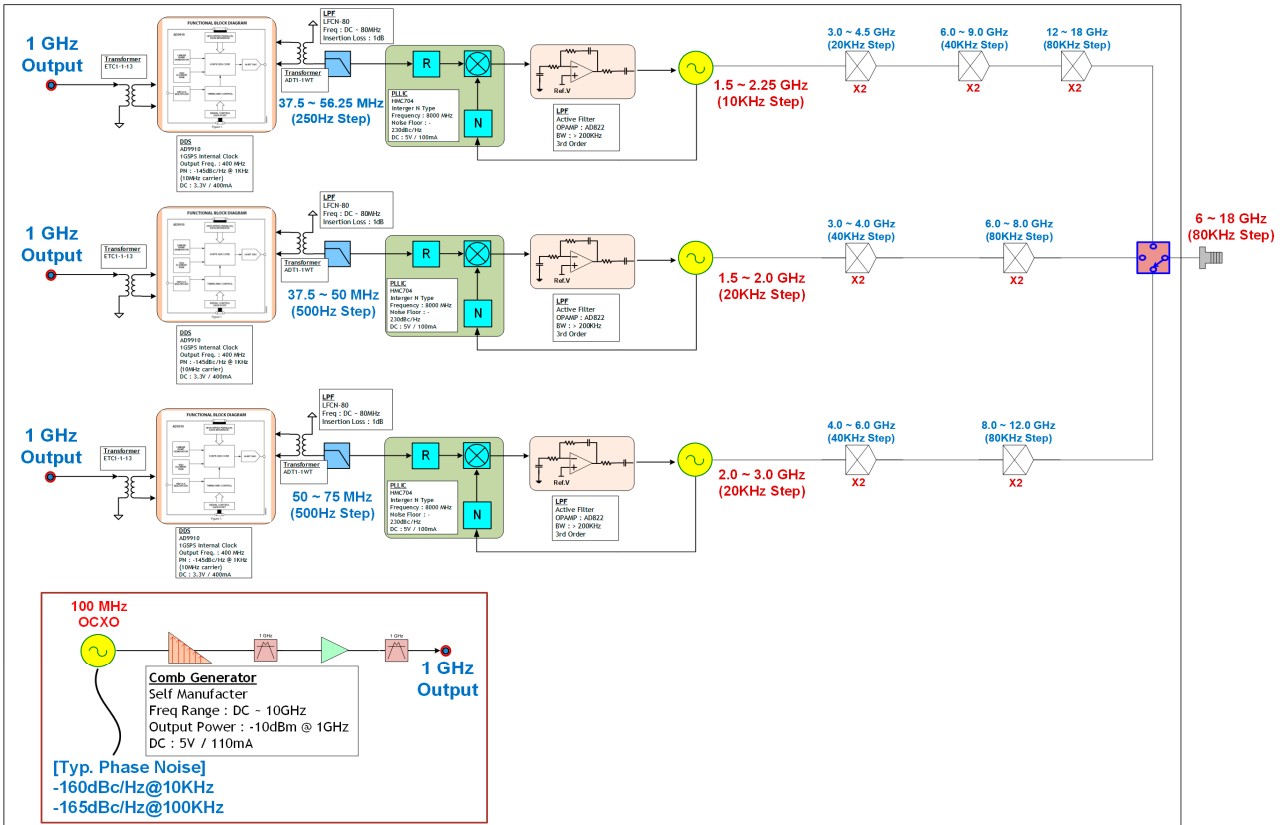

**Figure 2.** Block diagram of wideband frequency synthesizer with 6-18 GHz output based on PLVCO (old version).

### 2.1. Analysis of Phase Noise for Comb Generator

The comb generator structure with excellent phase noise characteristics and little difficulty in the implementation of the circuit was selected. The harmonic component of input signals is output, while the input signals pass through the step recovery diode (SRD) [11]. Table 1 shows that the phase noise of the N-multiplied frequency from an external reference at 1 GHz output will be deteriorated by as much as the multiplying factor as per Equation (1) below.

$$Phase\ Noise = 20 \times \log_{10}(N) \tag{1}$$

**Table 1.** Analyzed values of the phase noise (PN) for the comb generator method(dBc/Hz).

|  | Offset Frequency | | | | |
|---|---|---|---|---|---|
|  | **100 Hz** | **1 kHz** | **10 kHz** | **100 kHz** | **1 MHz** |
| P/N of external ref. | −130 | −155 | −160 | −165 | −168 |
| P/N after multiplier (×10) | −110 | −135 | −140 | −145 | −148 |

### 2.2. Considering for Active Loop Filter

The active loop filter has a wide frequency range of VCO for phase locking, so it needs to be raised to the corresponding voltage tune (VT) voltage (near +10 V). It should be increased at this time, since the Op-Amp needs VCC up to +10 V or more; pay special attention to noise so as not to affect the phase noise when supplying the corresponding noise of the power line and the corresponding power. Therefore, if possible, the PLL circuit using VCO components is not implemented in the frequency synthesizer module, which is sensitive to phase noise.

### 2.3. Analysis for Address Information of PLLIC

When setting the address value required to operate PLL IC parts, select parts with fractional function so that the integer (INT) value does not increase. Since the DDS output frequency can be used as the REF input frequency, change the phase frequency detector (PFD) value while maintaining the INT value (40). It is common to design in the form of giving. Table 2 shows the address setting values for one representative case with 2–3 GHz of PLVCO. In order to obtain the same phase noise within the VCO variable frequency (1.5–3.0 GHz), the INT value was maintained but the REF frequency (DDS output) was designed to be variable. The first and second VCOs were set to 20 KHz and the third VCO was set to a 20 KHz step so that the step size was changed to 80 KHz in the final output stage.

**Table 2.** Address values of PLLIC for 2.0–3.0 GHz (PLVCO).

| Fout (MHz) | PFD (MHz) | REF (MHz) | INT (20~524,283) | FRAC (1~16,777,216) | P (8) | R (1~16,384) | MOD (2~4095) |
|---|---|---|---|---|---|---|---|
| 2000.0000 | 50.00000 | 50.00000 | 40 | 0 | 8/9 | 1 | 2 |
| 2000.0200 | 50.00050 | 50.00050 | 40 | 0 | 8/9 | 1 | 2 |
| 2000.0400 | 50.00150 | 50.00150 | 40 | 0 | 8/9 | 1 | 2 |
| 2000.0600 | 50.00200 | 50.00200 | 40 | 0 | 8/9 | 1 | 2 |
| 2000.0800 | 50.00250 | 50.00250 | 40 | 0 | 8/9 | 1 | 2 |
| . | . | . | . | . | . | . | . |
| 2999.9800 | 74.99950 | 74.99950 | 40 | 0 | 8/9 | 1 | 2 |
| 3000.0000 | 75.00000 | 75.00000 | 40 | 0 | 8/9 | 1 | 2 |

*2.4. Analysis of the Phase Noise of the PLL Circuit*

The overall phase noise of the PLL is as follows:

$$\text{Total PN} = 10 \times \log\left(10^{\frac{VCO\ PN}{10}} + 10^{\frac{REF\ PN}{10}} + 10^{\frac{PS\ PN}{10}} + 10^{\frac{PD\ PN}{10}}\right) \tag{2}$$

The phase noise of the final output is calculated by Equation (2). The characteristics of the individual components are as follows: The phase noise of the VCO output under No. 1 is the value obtained after passing the high-pass attenuation of the passive loop filter. The phase noise of the reference output under No. 2 is the value obtained after an increase in the N divider ratio. This value is one (40) set in the PLLIC obtained by dividing the VCO frequency by the external reference frequency, followed by passing the low-pass attenuation through the passive loop filter. The phase noise of the pre-scaler output under No. 3 is a value obtained after passing the low-pass attenuation through the passive loop filter. The phase noise of the phase detector output under No. 4 can be calculated by Equation (3), followed by passing the low-pass attenuation through the passive loop filter.

$$\text{Close} - \text{in PN} = \text{PLLIC's noise floor} - 10 \times \log(\text{PFD}) - 20 \times \log(\text{N}), \tag{3}$$

where PFD is the phase frequency detector, and N is the divider and pre-scaler ratio of the digital PLLIC. Table 3 shows the final output frequency (6–18 GHz) for three VCOs including the multiplier path. In Table 3, the same characteristics can be obtained in the entire range of 6–18 GHz depending on how much the phase noise degradation characteristic is reduced due to the increase in the multiplication path.

**Table 3.** Summary of output frequency for three VCOs (GHz).

| VCO Freq. | Multiple (×2) | Multiple (×2) | Multiple (×2) | Total Output Freq. |
|-----------|---------------|---------------|---------------|--------------------|
| 1.5–2.0 | 3.0–4.0 | 6.0–8.0 | - | 6.0–8.0 |
| 2.0–3.0 | 4.0–6.0 | 8.0–12.0 | - | 8.0–12.0 |
| 1.5–2.25 | 3.0–4.5 | 6.0–9.0 | 12.0–18.0 | 12.0–18.0 |

Table 4 shows the final phase noise values of PLVCO (2.0–3.0 GHz) after multiplication (×4) calculations, where −100.0 dBc/Hz and −102.6 dBc/Hz are the offset frequencies at 1 kHz and 10 kHz for the 8~12 GHz output for one representative case. Table 5 shows the same step size in the final output frequency (6–18 GHz) for three VCOs including the multiplier, and it importantly operates as a local oscillator or BIT signal in the system level due to its equal step size.

**Table 4.** Phase noise calculation for 2.0–3.0 GHz (PLVCO).

| 2.0–3.0 GHz (PLVCO) | BW (kHz) | PFD (MHz) | FO (GHz) | N | | |
|---------------------|----------|-----------|----------|-----|-----|-----|
| 20 KHz step | 100 | 75 | 3.0 | 40 | | |
| Frequency [Hz]@OFFSET | 10 | 100 | 1000 | $1.0 \times 10^4$ | $1.0 \times 10^5$ | $1.0 \times 10^6$ |
| VCO phase noise | −20.0 | −45.0 | −78.0 | −102.0 | −123.0 | −142.0 |
| Reference phase noise | −88.0 | −123.0 | −145.0 | −150.0 | −160.0 | −165.0 |
| Pre-scaler phase noise | −135.0 | −140.0 | −145.0 | −148.0 | −148.0 | −148.0 |
| PD phase noise | −119.2 | −119.2 | −119.2 | −119.2 | −119.2 | −119.2 |
| High-pass att. (HPF) | 140.0 | 100.0 | 60.0 | 20.0 | 0.0 | 0.0 |
| Low-pass att. (LPF) | 0.0 | 0.0 | 0.0 | 0.0 | 0.0 | 20.0 |
| Reference multiply | −56.0 | −91.0 | −113.0 | −118.0 | −128.0 | −133.0 |
| No.1 VCO output | −160.0 | −145.0 | −138.0 | −122.0 | −123.0 | −142.0 |
| No.2 Reference output | −56.0 | −91.0 | −113.0 | −118.0 | −128.0 | −153.0 |

**Table 4.** *Cont.*

| 2.0–3.0 GHz (PLVCO) | BW (kHz) | PFD (MHz) | FO (GHz) | N | | |
|---|---|---|---|---|---|---|
| No.3 Pre-scaler output | −135.0 | −140.0 | −145.0 | −148.0 | −148.0 | −168.0 |
| No.4 Phase detector output | −119.2 | −119.2 | −119.2 | −119.2 | −119.2 | −139.2 |
| PLVCO's phase noise | −56.0 | −91.0 | −112.0 | −114.6 | −117.3 | −137.3 |
| 4–6 GHz PLVCO PN (X2) | −50.0 | −85.0 | −106.0 | −108.6 | −111.3 | −131.3 |
| 8–12 GHz PLVCO PN (X2) | −44.0 | −79.0 | −100.0 | −102.6 | −105.3 | −125.3 |

**Table 5.** Summary of step size for three VCOs.

| VCO Freq. (GHz) | DDS (Hz) | VCO (KHz) | X2 (KHz) | X2 (KHz) | X2 (KHz) |
|---|---|---|---|---|---|
| 1.5–2.0 | 500 | 20 | 40 | 80 | - |
| 2.0–3.0 | 500 | 20 | 40 | 80 | - |
| 1.5–2.25 | 250 | 10 | 20 | 40 | 80 |

## 3. Understanding of Wideband Frequency Synthesizer for 6–18 GHz Range Based on DDS (New version)

In this study, Figure 3 is a frequency synthesizer with an upgraded structure. In order to output 0.5–18 GHz from a single port, it has one DDS chip, a 3 GHz reference source for driving DDS using a multiplier path from a 100 MHz external source, and many different paths. The wideband synthesizer required in the electronic warfare field needs a frequency for LO because it has to down-convert the frequency and deliver it to the signal processing stage.

In addition, a separate frequency for BIT is required to input the same frequency to the BIT port of the first stage of the receiver module to determine whether the RF path is normal or not and for calibration before system installation [12].

The 3 GHz frequency, which generates the clock frequency of DDS, uses a multiplication method rather than a comb generator method to exclude other components other than the theoretically predictable deterioration of phase noise. Since there is no need for different VCOs and PLLICs, the frequency variable time of DDS is the same as the total locking time, so very fast response characteristics can be secured. Figure 3 shows a block diagram of a design with one DDS, a tunable filter, and a filter bank path.

### 3.1. Understanding How to Make Clock Signal for DDS Operation

It is very difficult to manufacture a reference signal source (100 MHz) with very good phase noise characteristics to increase the value by a theoretical multiple in an actual implementation circuit. In order to make a reference signal source of 1 GHz or higher for driving the DDS chip, the frequency is increased by using a 100 MHz OCXO with excellent frequency stability as a reference source. Therefore, although the circuit is rather complicated, only the phase noise deterioration by the theoretical N times is output, so the phase noise characteristics are excellent. In addition, the heterojunction bipolar transistor (HBT) component based on SiGe material was designed as a component having no deterioration in phase noise to compensate for the loss generated during the multiplication stage. Figure 4 presents how to design multiplier circuits to create a 3 GHz frequency and Table 6 shows the phase noise of the N-multiplied frequency from an external reference at 3 GHz output, which will be deteriorated by as much as the multiplying factor as per Equation (1). Figure 5a shows the value of phase noise at 100 KHz offset for the 100 MHz reference before multiplying, and Figure 5b,c present a photo of the evaluation board and results of the phase noise after multiplying (×10).

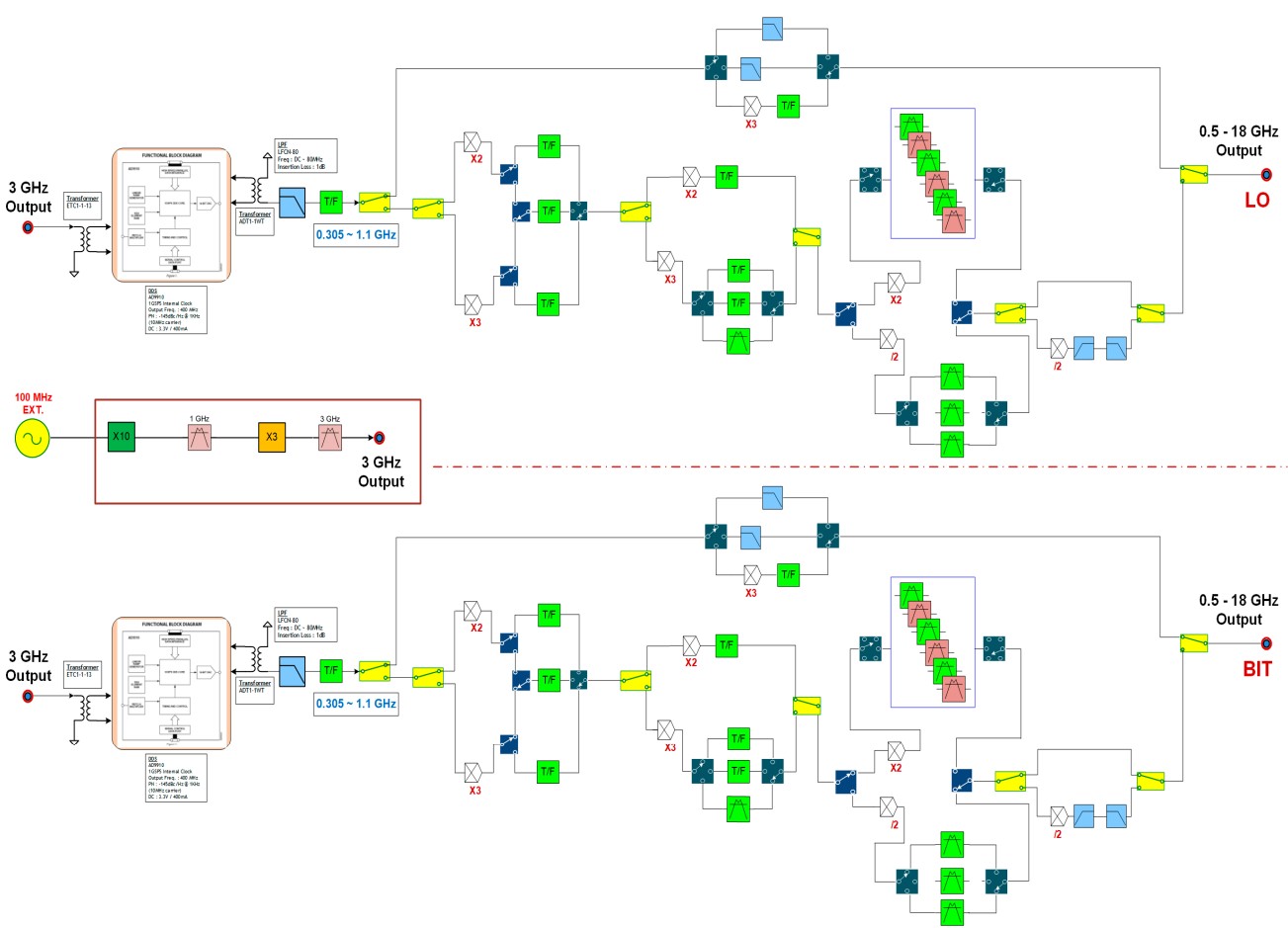

**Figure 3.** Block diagram of wideband frequency synthesizer with 0.5–18 GHz output based on DDS (new version).

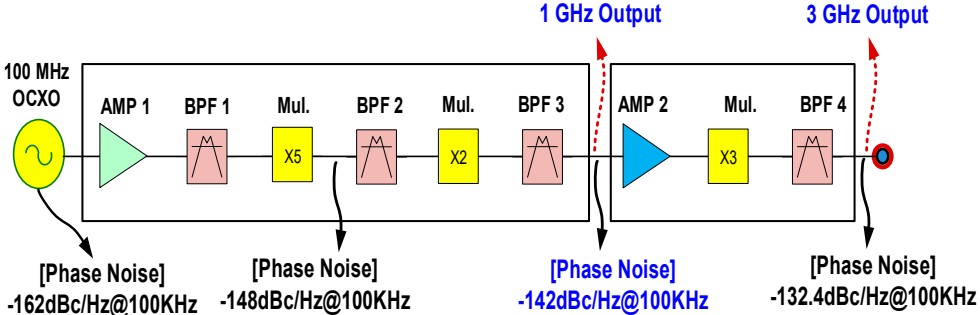

**Figure 4.** Block diagram of a 3 GHz output using the multiplier method.

**Table 6.** Analyzed values of the phase noise (PN) for the multiplier method (dBc/Hz).

|  | Offset Frequency | | | | |
|---|---|---|---|---|---|
|  | 100 Hz | 1 kHz | 10 kHz | 100 kHz | 1 MHz |
| P/N of external ref. | −130 | −155 | −160 | −162 | −168 |
| P/N after multiplier (×10) | −110 | −135 | −140 | −142 | −148 |
| P/N after multiplier (×3) | −100.5 | −125.5 | −130.5 | −132.5 | −138.5 |

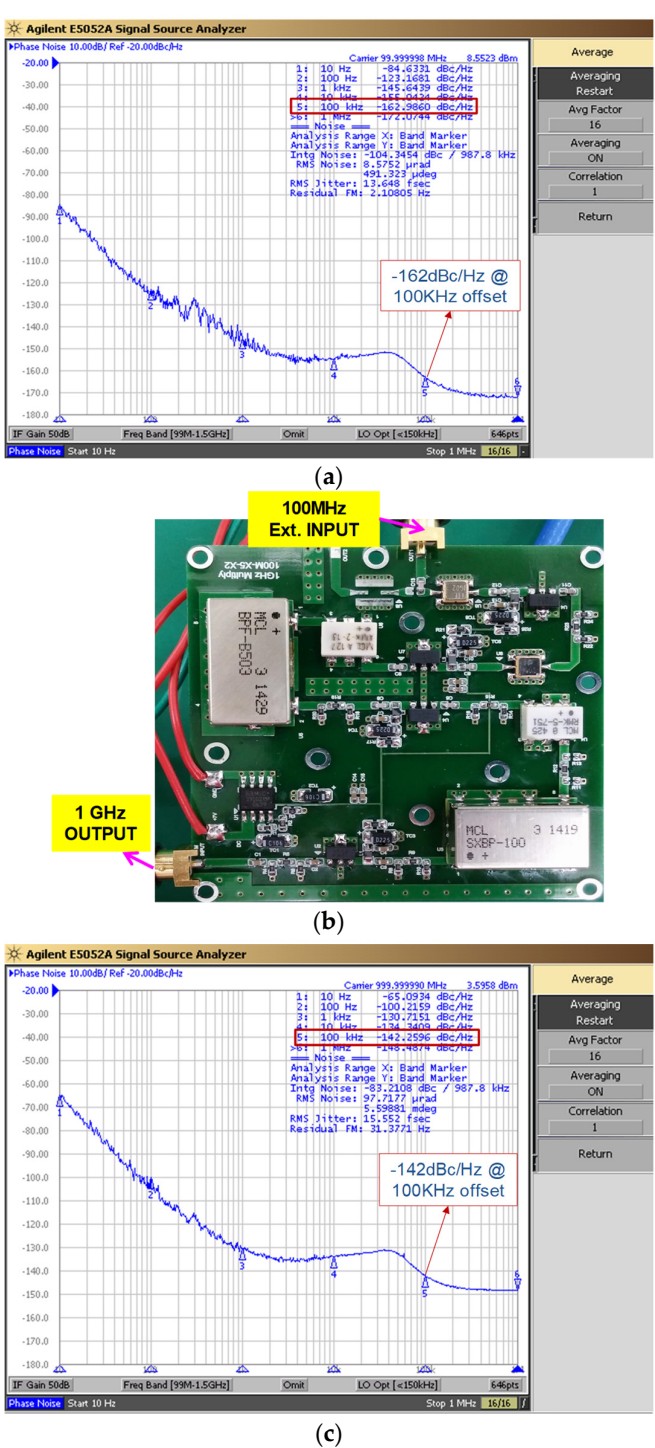

**Figure 5.** (**a**) The results of phase noise at 100 KHz offset before multiplying for 100 MHz. (**b**) The photos of evaluation board for 1 GHz output. (**c**) The results of phase noise at 100 KHz offset after multiplying for 1 GHz.

The results are the same as the analyzed values at 100 KHz offset (values in bold font in Table 6), but −132.5 dBc/Hz is the only expected value because it was not generated.

## 3.2. Analysis and Implementation for DDS

As an abbreviation of "direct digital synthesizer", DDS is the principle of outputting frequency using only the external clock frequency. The maximum output frequency can use up to half of the external clock frequency. For example, if the external clock frequency

is 3 GHz, the maximum is 1.5 GHz. However, since 1.5 GHz, which is half of the clock frequency, is output simultaneously at the output frequency, the maximum value that the filter can suppress must be designed (considering at least several hundred MHz lower to easily reject). Figure 6 shows the general structure of DDS and how to create the desired frequency.

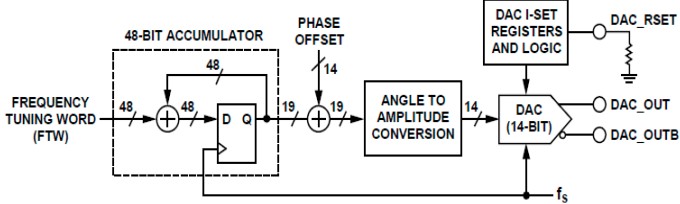

**Figure 6.** The simple block diagram of DDS chip for ADI manufacture [13].

In the DDS, the close-in phase noise is determined by the purity of the clock source. The DDS divides the clock frequency by some real number. Therefore, the close-in phase noise is reduced by $20 \times \log_{10}(N)$ (2.4), where N (N = Fclk/Fout) is a division ratio between the DDS clock and the output frequency in Figure 7. Of course, the DDS circuitry has a noise floor that, at some point, will limit this improvement [14].

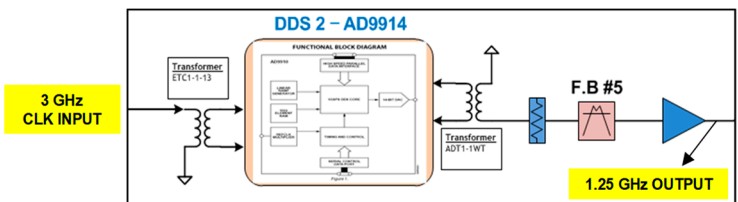

**Figure 7.** Block diagram of 3 GHz output using DDS.

Table 7 shows the different value of phase noise theory value and measured value for the external clock source with 3 GHz input. It shows that the measured value does not improve as much as the theoretical value (N: 2.4, 7.6 dB) at all offsets.

**Table 7.** The values of the phase noise (PN) for the DDS's input vs. output (dBc/Hz).

|  | Offset Frequency | | | | |
|---|---|---|---|---|---|
|  | **100 Hz** | **1 kHz** | **10 kHz** | **100 kHz** | **1 MHz** |
| P/N of 3.0 GHz (DDS clock) | −95 | −118 | −129 | −132 | −133 |
| P/N of 1.25 GHz (Theory value) | −103 | −126 | −137 | −140 | −141 |
| P/N of 1.25 GHz (Measured value) | −101 | −122 | −131 | −136 | −138 |

Figure 8a shows the evaluation board, including the DDS chip with 3 GHz clock frequency, and Figure 8b,c present the value of the phase noise for the 3 GHz clock source and 1.25 GHz output frequency.

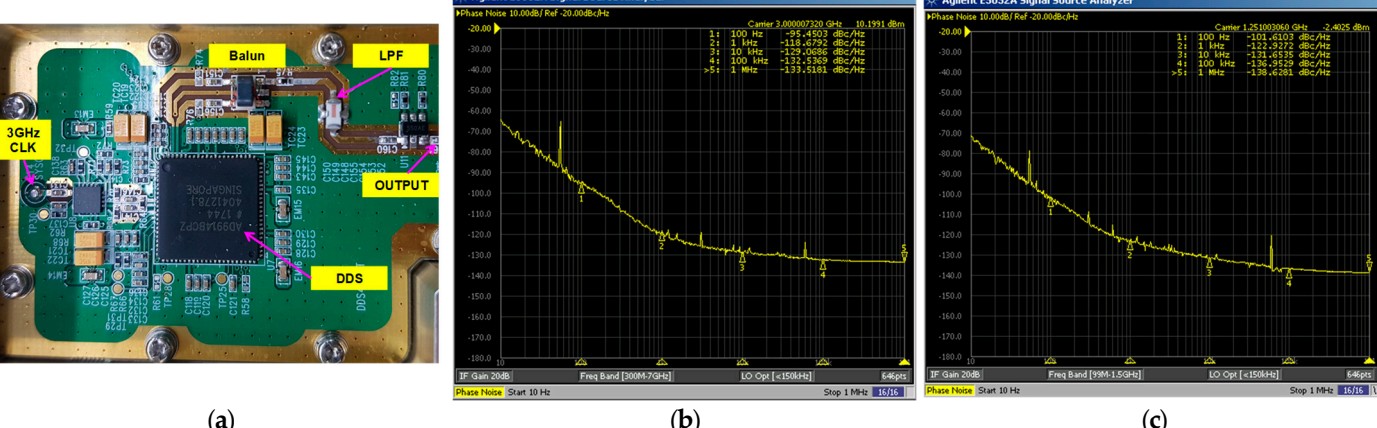

|  |  |  |
|:--:|:--:|:--:|
| (**a**) | (**b**) | (**c**) |

**Figure 8.** Manufactured and measured data: (**a**) photo of the evaluation board including DDS chip; (**b**) the results of phase noise for 3 GHz clock source; (**c**) the results of phase noise for 1.25 GHz output.

### 3.3. Understanding How to Pass Wanted Signal Band Using Digitally Tunable Filter

Based on Figure 3, the frequency range required for a tunable filter is from 0.75 to 6.0 GHz after DDS output, and the range of the selected tunable filter (AM3060) satisfies this operating frequency. The low frequency and narrow bandwidth of the signal source (DDS output) is converted to a high frequency and very wide bandwidth through a multiplier, divider, and filter bank area. In this case, in order for the filter bank to pass only a desired band while simultaneously satisfying the suppression characteristics of adjacent frequencies, it is essential to use a tunable filter instead of a fixed band-pass filter that is bulky and has poor suppression characteristics for other frequencies. The advantage of this is that the frequency can be easily changed and only a desired signal can be output according to the operating scenario frequency with one component (chip). Figure 9 is an internal configuration diagram of the tunable filter, and the output is configured to obtain band-pass filter (BPF) characteristics composed of high-pass filters (HPFs) and low-pass filters (LPFs) for each different frequency path.

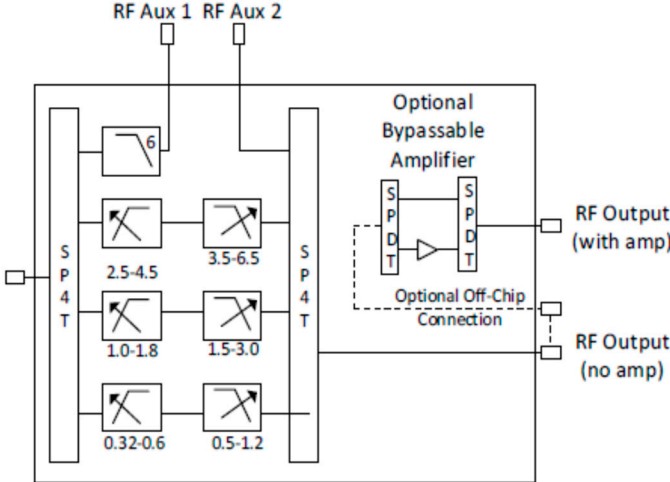

**Figure 9.** The results of phase noise for 1.25 GHz output [15].

Figure 10a–c present the values of passband frequency for each path with rejection values. Table 8 shows the summarized values. In the case of the 2.5–6.5 GHz range, the high side rejection is worse than in other bands.

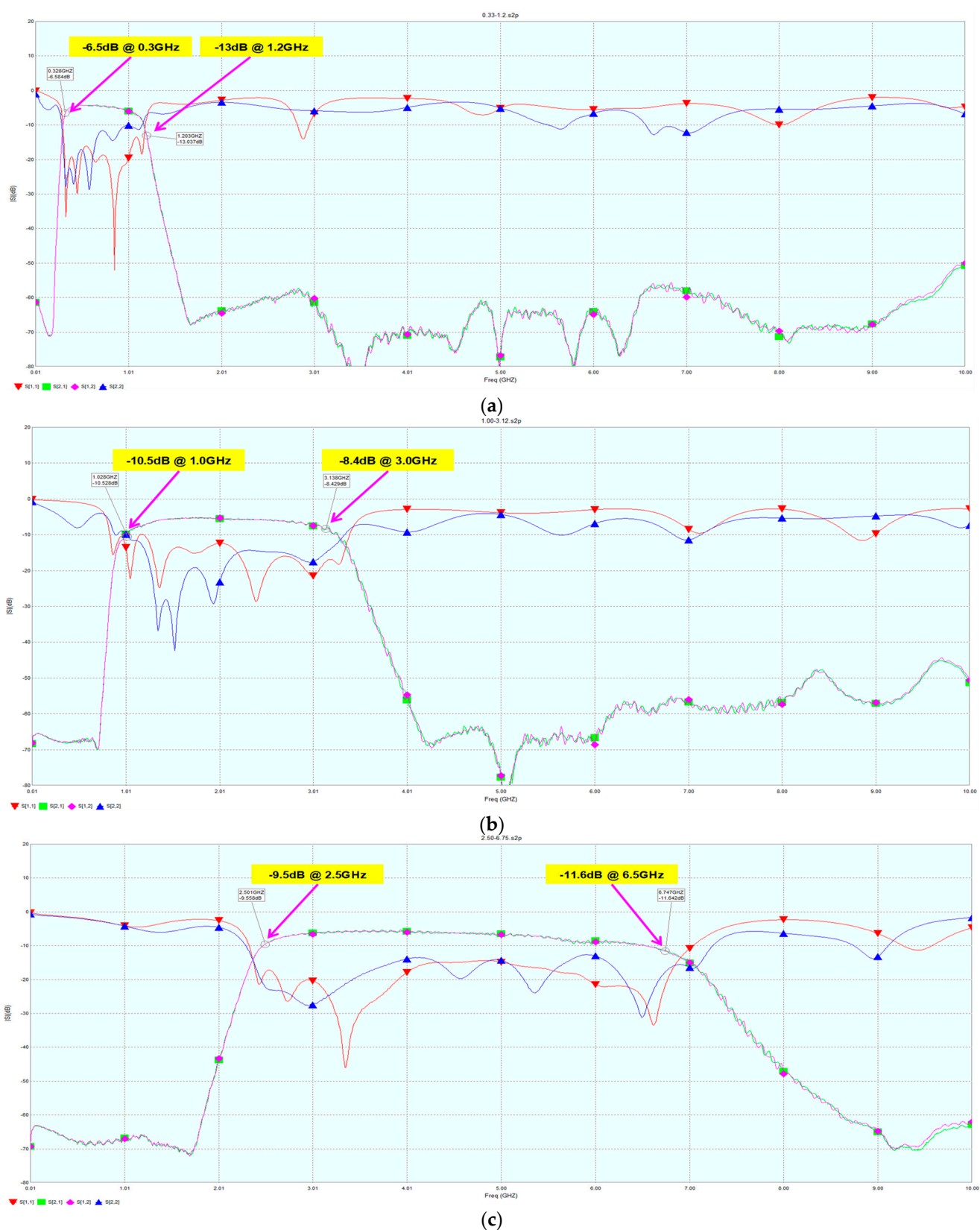

**Figure 10.** (**a**) The measured graph for 0.32–1.2 GHz path. (**b**) The measured graph for 1.0–3.0 GHz path. (**c**) The measured graph for 2.5–6.5 GHz path.

**Table 8.** The values of the insertion loss and rejection for the measured tunable filter.

| BPF Band | Insertion Loss (dB) | | Rejection (dBc) | |
|---|---|---|---|---|
| 0.32–1.2 GHz | −6.5 | −13.0 | 60 @ 0.2 GHz | 60 @ 1.7 GHz |
| 1.0–3.0 GHz | −10.5 | −8.4 | 60 @ 0.8 GHz | 60 @ 4.2 GHz |
| 2.5–6.5 GHz | −9.5 | −11.6 | 60 @ 1.7 GHz | 35 @ 8.0 GHz |

### 3.4. Analysis How to Reject for Spurious Signals within Operating Frequency Band

The most important point in the frequency synthesizer is whether to remove unnecessary waves other than the desired output signal. Since unwanted waves are generated from one signal source (DDS output) through multiple stages such as bypass, multiplier, and divider, it is necessary to establish a frequency plan for each path so that it can be suppressed by a filter. In addition, when the output frequency is generated, the original signal and other multiplication components must be removed after the multiplication stage because it is implemented only as a multiplier as it goes up to a higher frequency.

Table 8 shows through which path the 0.5–18 GHz output frequency composed of 12 paths is obtained. For each path, it is necessary to check whether an unwanted signal exists within the band or whether suppression is possible using a filter when an adjacent frequency exists outside the band. Figure 11a, b show frequency plans for each path based on Table 9, which shoes how to reject unwanted spurious signals.

**Table 9.** The analysis for output frequency using each paths.

| No. | Output of DDS (GHz) | | Multiplier Path (GHz)/Output Frequency (0.5–18 GHz) | | | | | | | |
|---|---|---|---|---|---|---|---|---|---|---|
| | | | Bypass | | | | | | | |
| 1 | 0.5 | 0.7 | 0.5 | 0.7 | | | | | | |
| | | | Bypass | | | | | | | |
| 2 | 0.7 | 1.1 | 0.7 | 1.1 | | | | | | |
| | | | X2 | | X3 | | /2 | | /2 | |
| 3 | 0.733 | 1.0 | 1.466 | 2.0 | 4.398 | 6.0 | 2.199 | 3.0 | 1.10 | 1.5 |
| | | | Bypass | | X3 | | | | | |
| 4 | 0.5 | 0.65 | 0.5 | 0.65 | 1.5 | 1.95 | | | | |
| | | | X2 | | X3 | | /2 | | | |
| 5 | 0.65 | 0.8 | 1.3 | 1.6 | 3.9 | 4.8 | 1.95 | 2.4 | | |
| | | | X3 | | X3 | | /2 | | | |
| 6 | 0.533 | 0.6666 | 1.599 | 1.9998 | 4.797 | 5.9994 | 2.4 | 2.9997 | | |
| | | | X2 | | X2 | | X2 | | | |
| 7 | 0.375 | 0.5625 | 0.75 | 1.125 | 1.5 | 2.25 | 3.0 | 4.5 | | |
| | | | X2 | | X3 | | X2 | | | |
| 8 | 0.375 | 0.4583 | 0.75 | 0.9166 | 2.25 | 2.7498 | 4.50 | 5.4996 | | |
| | | | X3 | | X3 | | X2 | | | |
| 9 | 0.305 | 0.4 | 0.915 | 1.2 | 2.745 | 3.6 | 5.49 | 7.2 | | |
| | | | X3 | | X3 | | X2 | | | |
| 10 | 0.4 | 0.5 | 1.2 | 1.5 | 3.6 | 4.5 | 7.2 | 9.0 | | |

**Table 9.** *Cont.*

| No. | Output of DDS (GHz) | | Multiplier Path (GHz)/Output Frequency (0.5–18 GHz) | | | | | |
|---|---|---|---|---|---|---|---|---|
| | | | X3 | | X3 | | X2 | |
| 11 | 0.5 | 0.6667 | 1.5 | 2.0001 | 4.5 | 6.0003 | 9.0 | 12.0006 |
| | | | X3 | | X3 | | X2 | |
| 12 | 0.6667 | 1.0 | 2.0001 | 3.0 | 6.0003 | 9.0 | 12.0006 | 18.0 |

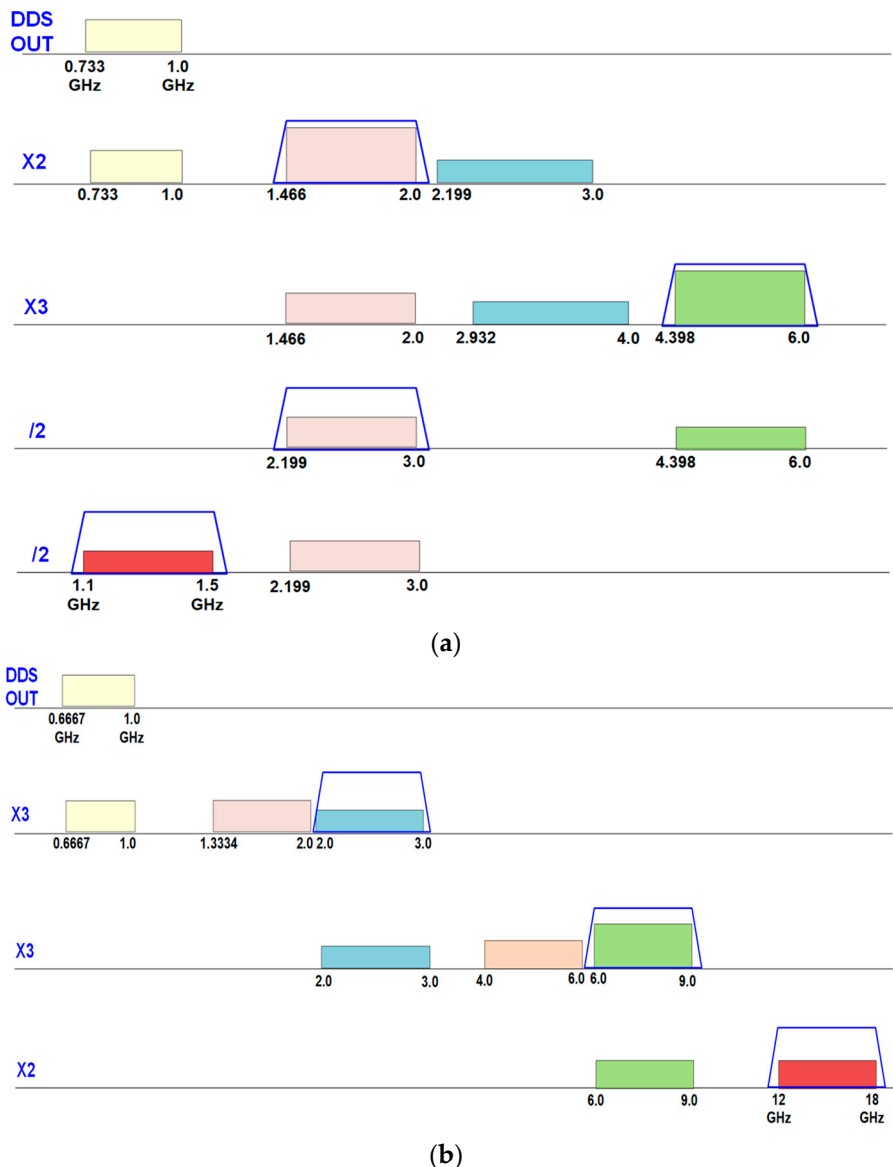

**Figure 11.** (**a**) The frequency plan for No.3 path in Table 9. (**b**) The frequency plan for No.12 path in Table 9.

### 3.5. Analysis of the Phase Noise in the Final Output

Based on the frequency plan configured as shown in Table 9, the results of the phase noise characteristic analysis at the final output are shown in Table 10. At a low frequency, the phase noise characteristic is good because there are relatively few multiplier stages, but at a high frequency of 3 GHz or more, the phase noise characteristic is not good because the output must be obtained through 18 multiplying stages. However, even considering the worst case, it is analyzed that the characteristic of −105.9 dBc at 1 KHz offset can be

obtained. This is an excellent noise characteristic of a wideband frequency synthesizer, and it is judged that it has no problem with serving as LO and BIT signals.

**Table 10.** Phase noise calculation for the proposed synthesizer (new version).

| Frequency [Hz]@OFFSET | DDS | Phase Noise for Each Path | | | | Total P/N (dBc/Hz) | Freq. Output [GHz] |
|---|---|---|---|---|---|---|---|
| | | X2 | X3 | /2 | /2 | | |
| 100 | −101.0 | | | | | −101.0 | |
| 1000 | −122.0 | | | | | −122.0 | |
| $1.0 \times 10^4$ | −131.0 | | | | | −131.0 | 0.5–1.1 |
| $1.0 \times 10^5$ | −136.0 | | | | | −136.0 | |
| $1.0 \times 10^6$ | −138.0 | | | | | −138.0 | |
| 100 | −101.0 | −95.0 | −85.4 | −91.5 | −97.5 | −97.5 | |
| 1000 | −122.0 | −116.0 | −106.4 | −112.5 | −118.5 | −118.5 | |
| $1.0 \times 10^4$ | −131.0 | −125.0 | −115.4 | −121.5 | −127.5 | −127.5 | 1.1–1.5 |
| $1.0 \times 10^5$ | −136.0 | −130.0 | −120.4 | −126.5 | −132.5 | −132.5 | |
| $1.0 \times 10^6$ | −138.0 | −132.0 | −122.4 | −128.5 | −134.5 | −134.5 | |
| 100 | −101.0 | −95.0 | | | | −95.0 | |
| 1000 | −122.0 | −116.0 | | | | −116.0 | |
| $1.0 \times 10^4$ | −131.0 | −125.0 | | | | −125.0 | 1.5–1.95 |
| $1.0 \times 10^5$ | −136.0 | −130.0 | | | | −130.0 | |
| $1.0 \times 10^6$ | −138.0 | −132.0 | | | | −132.0 | |
| 100 | −101.0 | −95.0 | −85.4 | −91.5 | | −91.5 | |
| 1000 | −122.0 | −116.0 | −106.4 | −112.5 | | −112.5 | |
| $1.0 \times 10^4$ | −131.0 | −125.0 | −115.4 | −121.5 | | −121.5 | 1.95–3.0 |
| $1.0 \times 10^5$ | −136.0 | −130.0 | −120.4 | −126.5 | | −126.5 | |
| $1.0 \times 10^6$ | −138.0 | −132.0 | −122.4 | −128.5 | | −128.5 | |
| | DDS | X3 | X3 | X2 | | | |
| 100 | −101.0 | −91.5 | −81.9 | −75.9 | | −75.9 | |
| 1000 | −122.0 | −112.5 | −102.9 | −96.9 | | −96.9 | |
| $1.0 \times 10^4$ | −131.0 | −121.5 | −111.9 | −105.9 | | −105.9 | 3.0–18.0 |
| $1.0 \times 10^5$ | −136.0 | −126.5 | −116.9 | −110.9 | | −110.9 | |
| $1.0 \times 10^6$ | −138.0 | −128.5 | −118.9 | −112.9 | | −112.9 | |

Table 11 shows the comparison table for the main performance such as phase noise, frequency range, and output port quantity.

**Table 11.** The comparison table among old vs. new type.

| | P/N (dBc/Hz) | | Freq. Range (GHz) | Output (EA) |
|---|---|---|---|---|
| VCO Type (OLD) | −100 @ 1 KHz offset | −102.6 @ 10 KHz offset | 6–18 | 1 (LO) |
| DDS Type (NEW) | −105.9 @ 1 KHz offset | −110.9 @ 10 KHz offset | 0.5–18 | 2 (LO/BIT) |

## 4. Conclusions

In the present study, a new structure (0.5–18 GHz) with excellent phase noise characteristics is proposed. Ultra-wideband synthesizers with low phase noises and less spurious signals have been developed to be used as local oscillators and for built-in test (BIT) functions in the field of electronic warfare system (EW), in which synthesizers are installed at the front-end of devices. I tried to reduce the risk factor in the actual design stage by identifying and analyzing the shortcomings of the old version's (6–18 GHz) structure and supplementing and reflecting it in the new version (0.5–18 GHz) structure. In addition, by

securing two ports of the same performance, I tried to take advantage of the ability to provide LO and BIT signals at the same time. At this time, different frequencies can be output from the two ports using different FPGA coding of the internal frequency plan according to the operating frequency. It is hoped that this study will help engineers who want to design a wideband frequency synthesizer to make a product with better cost-effectiveness by identifying, in advance, this problem that is small in volume and unexpected. The advantages of the proposed new structure are as follows:

1. Prevent PLL unlock for multiple oscillator circuits.
2. Output the DDS output signal so that the secondary harmonic component does not occur.
3. Because there is only one source, predictable phase noise analysis is possible.
4. Frequency plan design so that unwanted signals can be easily suppressed by applying filters to each path.
5. Applicable as a T/F part without the need for a special type of filter in some routes.
6. PCB one-board implementation and SMT are possible using package parts.
7. Due to No. 6, the assembly time is shortened, the price is reduced, and commercialization is easy.
8. The switching speed is very fast because the frequency is changed at the DDS output without going through a separate PLL circuit.

In particular, by proposing a structure for obtaining a wideband frequency using a single source (DDS), it can be said that it is a structure that secures reliability from the point of view of a system operating for a long time by implementing a similar circuit within a predictable range. This study will help students and engineers who want to design a wideband frequency synthesizer to make a product with better cost-effectiveness by identifying, in advance, this problem that is small in volume and unexpected.

**Funding:** This research received no external funding.

**Data Availability Statement:** Data are contained within the article.

**Conflicts of Interest:** The author was employed by the company Broadern Inc. The author declares that the research was conducted in the absence of any commercial or financial relationships that could be construed as a potential conflict of interest.

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
