# Peer review of "Study of Ultra-Broadband Synthesizer of Fast Indirect Type in a 0.5–18 GHz Range for SIGINT System"

_electronics, doi:10.3390/electronics13010048_

Round 1
Reviewer 1 Report
Comments and Suggestions for Authors
The main question addressed by the research is novel realization of broadband synthesizer for the frequency range from 0.5 GHz to 18 GHz.
The paper topic is original in the field of microwave synthesizer realizations. The existing configuration for 6-18 GHz is extended to 0.5 GHz.
This research addresses a few specific contributions in comparison with published papers such as: prevent PLL unlock for multiple oscillators; the secondary harmonic is suppressed; predictability of phase noise analysis; unwanted signals suppressions by applying filters to each path; applicability as a T/F part, single laminate implementation which is easier and cheaper; faster switching speed.
The authors should consider enclosing (housing) the proposed synthesizer realization in contest of electromagnetic compatibility i.e. EM leakage.
The conclusions consistent with the evidence and arguments presented and they address the main research question.
All listed references in the paper are appropriate, but they are not presented in the MDPI template.
Additional comments on the tables and figures are such as following: in the text “Table” and “Figure” should be written with the first (initial) uppercase letter.
Author Response
Thank you for kindly review about my manuscript.
PLS see the attached file including revised sentence and reference template.

Reviewer 2 Report
Comments and Suggestions for Authors
What specific improvements should the authors consider regarding the methodology?
For a better understanding of the material, I would suggest the Authors would address the following MINOR points:
A. Fig. 2: the insets in the figure are readable with very high difficulty. The Authors are invited to make them larger and/or more readable.
B. B. Fig. 5: what are the technical characteristics/properties of the probe used to measure the phase noise shown in these Figure 5?
Author Response
Thank you for kindly review about my manuscript.
Figure 2 was revised from old to new which was larged font size.
Figure 5 does not need the probe cable since phase noise was easily able to test because each port for 100MHz and 1GHz was made with SMA.

Reviewer 3 Report
Comments and Suggestions for Authors
This paper studies the "Study of Ultra Broadband Synthesizer for a 0.5-18GHz with Fast Indirect Type for SIGINT system".
The English need a few improvements.
1. The scientific sounds are good and explain very well with Figures.
2. All the cited references are relevant and describe the state of the art.
3. The novelty of the manuscript is high, with excellent phase noise characteristics and fast switching speed described very well.
4. The results are very well presented, with many explanations.
5. The conclusions need some improvement.
I recommand minor revison.
Comments on the Quality of English Language
The English need some improvement.
Author Response
Thank you for kindly review about my manuscript.
